# miR-98 Regulates TMPRSS2 Expression in Human Endothelial Cells: Key Implications for COVID-19

**DOI:** 10.3390/biomedicines8110462

**Published:** 2020-10-30

**Authors:** Alessandro Matarese, Jessica Gambardella, Celestino Sardu, Gaetano Santulli

**Affiliations:** 1Department of Medicine, Wilf Family Cardiovascular Research Institute, Einstein-Institute for Aging Research, Albert Einstein College of Medicine, New York, NY 10461, USA; alessandromatarese@yahoo.it (A.M.); jessica.gambardella@einsteinmed.org (J.G.); 2AORN “Antonio Cardarelli”, 80100 Naples, Italy; 3Department of Advanced Biomedical Science, “Federico II” University, and International Translational Research and Medical Education Consortium (ITME), 80131 Naples, Italy; 4Department of Molecular Pharmacology, Fleischer Institute for Diabetes and Metabolism (FIDAM), Einstein-Mount Sinai Diabetes Research Center (ES-DRC), Albert Einstein College of Medicine, New York, NY 10461, USA; 5Department of Advanced Medical and Surgical Sciences, University of Campania “Luigi Vanvitelli”, 80138 Naples, Italy; celestino.sardu@unicampania.it; 6Department of Medical Sciences, International University of Health and Medical Sciences “S. Camillo”, 00131 Rome, Italy

**Keywords:** ACE2, coronavirus, COVID-19, endothelium, epigenetics, HMVEC-L, HUVEC, lung, microRNA, miR-98-5p, non-coding RNA, SARS-CoV-2

## Abstract

The two main co-factors needed by the severe acute respiratory syndrome coronavirus 2 (SARS-CoV-2) to enter human cells are angiotensin-converting enzyme 2 (ACE2) and transmembrane protease serine 2 (TMPRSS2). Here, we focused on the study of microRNAs that specifically target TMPRSS2. Through a bioinformatic approach, we identified miR-98-5p as a suitable candidate. Since we and others have shown that endothelial cells play a pivotal role in the pathogenesis of the coronavirus disease 2019 (COVID-19), we mechanistically validated miR-98-5p as a regulator of TMPRSS2 transcription in two different human endothelial cell types, derived from the lung and from the umbilical vein. Taken together, our findings indicate that TMPRSS2 represents a valid target in COVID-19 treatment, which may be achieved by specific non-coding-RNA approaches.

## 1. Introduction

Caused by the severe acute respiratory syndrome coronavirus 2 (SARS-CoV-2), coronavirus disease 2019 (COVID-19) represents a dramatic public health crisis of global proportions, with more than 30 million documented infections and 1 million deaths worldwide [1,2,3,4,5]. One of the main co-factors needed by SARS-CoV-2 to access human host cells is represented by the cellular surface protein known as angiotensin-converting enzyme 2 (ACE2) [6,7,8]. However, the internalization of the coronavirus requires not only binding to ACE2 but also priming of the viral spike protein by the transmembrane protease serine 2 (TMPRSS2) [9,10]. Such a cleavage step is necessary for the actual virus–host cell membrane fusion and the subsequent cell entry [11,12].

Emerging evidence has shown that SARS-CoV-2 can directly target endothelial cells [13,14,15,16], an aspect initially suggested by the systemic manifestations observed in COVID-19 patients, recently confirmed by autoptic findings in different organs, including lung, gut, kidney, and heart [17,18,19,20,21]. Of note, whereas the role of ACE2 in endothelial function and in the pathogenesis of COVID-19 has been extensively investigated [22,23,24,25,26,27,28,29,30], the potential contribution of TMPRSS2 and its targeting as a novel therapeutic approach has been less studied.

MicroRNAs (miRNAs, miRs) are small, highly conserved, non-coding single-stranded ribonucleic acids (RNAs), which enhance messenger RNA (mRNA) degradation and/or inhibit protein translation by binding to the 3′- untranslated regions (UTRs) of target mRNAs [31,32,33,34]. They play essential regulatory roles in a number of biological processes, both in health and in disease, and have been extensively investigated in cardiovascular medicine [35,36,37,38]. Specifically, we and others have identified a number of miRNAs involved in the regulation of endothelial function [39,40,41,42,43,44]. In terms of therapeutic potential, miRNAs represent a very appealing strategy to manipulate various processes as their activity can be efficiently modulated with innovative RNA-based technologies [45,46].

The aim of this study was to identify and validate miRNAs that specifically target TMPRSS2 in human endothelial cells. A bioinformatic screening resulted in the identification of hsa-miR-98-5p as a highly conserved miRNA potentially capable of repressing TMPRSS2 mRNA expression. The mechanistic role of miR-98-5p was then validated by assessing the regulation of TMPRSS2 transcription levels in two different clones of human endothelial cells.

## 2. Materials and Methods

### 2.1. Cell Culture and Reagents

Adult human lung microvascular endothelial cells (HMVEC-L, Lonza, Basel, Switzerland; Catalog number: CC-2527) and human umbilical vein endothelial cells (HUVEC, ThermoFisher Scientific, Waltham, MA: Catalog number: #C0035C) were cultured in a 5% CO_2_ humidified atmosphere (37 °C). All reagents were from Millipore-Sigma (Burlington, MA, USA), unless otherwise stated.

### 2.2. Identification of miR-98-5p as a Regulator of TMPRSS2

To identify miRNAs targeting the 3′-UTR of TMPRSS2, we used online target prediction tools, including Targetscan version 7.2 and miRWalk-3, as we previously described [44,47,48,49]. These programs predict biological targets of miRNAs by searching for conserved sites that match the seed region of miRNAs.

### 2.3. Biological Validation of miR-98-5p as a Regulator of TMPRSS2

To assess the actual effects of miR-98-5p on TMPRSS2 gene transcription, we used a luciferase reporter containing the 3′-UTR of the predicted miRNA interaction sites, both wild-type and mutated, in HMVEC-L cells. 

The mutant construct of TMPRSS2 3′-UTR, carrying a substitution of two nucleotides within the predicted miR-98-5p binding sites of TMPRSS2 3′-UTR (Figure 1) was obtained using a site-directed mutagenesis kit (New England Biolabs, Ipswich, MA, USA), as described [47]. Using Lipofectamine RNAiMAX (ThermoFisher Scientific, Waltham MA, USA), cells were transfected with the 3′-UTR reporter plasmid (0.05 μg) and miR-98-5p mirVana^TM^ (50 nM) mimics or inhibitors (ThermoFisher Scientific) as well as a non-targeting negative control (scramble), according to the manufacturer’s instructions [47]. Forty-eight hours after transfection, Firefly and Renilla luciferase activities were assessed using a commercially available Luciferase Reporter Assay System (Promega, Madison, WI, USA). Firefly luciferase was normalized to Renilla luciferase activity. 

Levels of miR-98 were measured using TaqMan miRNA assay, according to the manufacturer’s instructions, normalizing the miR expression to the level of U6; standard TaqMan gene expression assays from Applied Biosystem were used, as we described [44,47,48,49]. Cellular mRNA expression of TMPRSS2 was determined by RT-qPCR, as we previously described [44,47,48,49], normalizing to endogenous glyceraldehyde 3-phosphate dehydrogenase (GAPDH).

Sequences of oligonucleotide primers (Merck KGaA, Darmstadt, Germany) are presented in Appendix A.

### 2.4. Immunoblotting 

Immunoblotting assays were performed as previously described by our group [44,47,49,50,51] and developed with the Odyssey system (LI-COR Biosciences, Lincoln, NE, USA). The intensity of the bands was quantified using the FIJI (“Fiji Is Just ImageJ”) software. 

The antibody for TMPRSS2 was purchased from ThermoFisher Scientific (catalog number: #MA5-35756); the antibody for β Actin was purchased from abcam (Cambridge, MA, catalog number: #ab8229).

### 2.5. Statistical Analysis

Data are expressed as means ± standard error of means (SEM). Statistical analyses were performed in Prism (GraphPad Software, Version 8.0; Prism, San Diego, CA, USA).

Statistical significance was tested using the nonparametric Mann–Whitney U test or two-way ANOVA followed by Tukey–Kramer multiple comparison test, as appropriate. Significant differences were established at *p* < 0.05.

## 3. Results

### 3.1. Identification of miR-98-5p as a Specific Modulator of TMPRSS2

Through bioinformatic analyses, we identified miR-98-5p as a potential regulator of TMPRSS2. The other top-ranked miRNAs identified by bioinformatic analysis were miR-4500 and miR-4458. However, we did not find any literature on the role of miR-4500 and miR-4458 in endothelial dysfunction, whereas recent reports [52,53,54] have suggested a role for miR-98-5p in endothelial cells. Moreover, the complementary nucleotides between the target region of TMPRSS2 3′ untranslated region (3′-UTR) and miR-98-5p are evolutionarily highly conserved across different species, including humans, non-human primates, and rodents (Figure 1).

### 3.2. TMPRSS2 Is a Molecular Target of miR-98-5p

Previous reports have demonstrated that human endothelial cells express miR-98-5p in basal conditions, and such expression has been shown to be modulated by different stimuli, including hypoxia [52] and oxidized low-density lipoprotein (LDL) [54].

The proposed relationship was substantiated by an actual validation of seed complementarity, confirming the interaction between miR-98-5p and TMPRSS2 3′-UTR both in HMVEC-L (Figure 2A) and HUVEC (Figure 2B) through a luciferase assay.

### 3.3. miR-98-5p Regulates TMPRSS2 Transcription Levels in Human Endothelial Cells

After having validated that miR-98-5p targets TMPRSS2 3′UTR, we verified the effects of miR-98-5p mimic and miR-98-5p inhibitor on the transcription levels of TMPRSS2 both in HMVEC-L (Figure 3A) and HUVEC (Figure 3B). These findings were also confirmed in terms of protein levels (Appendix A).

## 4. Discussion

In the present study, we demonstrate for the first time that miR-98-5p directly targets the 3′UTR of TMPRSS2. To our knowledge, we also provide the first evidence of the actual expression of TMPRSS2 in human endothelial cells, confirmed in endothelial cell lines obtained from different tissues such as the lung and the umbilical vein.

Our results are consistent with the evidence of a key role for miR-98 in the regulation of endothelial function. Indeed, miR-98 has been recently shown to reduce endothelial dysfunction by protecting the blood–brain barrier and improving neurological outcomes in ischemia/reperfusion [55]. Equally important, previous in vitro observations in HUVEC had linked miR-98 to an inhibition of the apoptotic effects of hypoxia/re-oxygenation [52] and of oxidized LDL [54].

Our data are significant in the clinical scenario; indeed, since TMPRSS2 is known to play a crucial role in the pathophysiology of COVID-19, our results could open the field to new research in order to verify the role of miR-98-5p in other tissues and cell types. It is important to note that several potential therapeutic strategies targeting ACE2 have been already proposed to tackle COVID-19; however, mainly owing to the critical metabolic and hemodynamic roles of ACE2, including the regulation of glucose homeostasis [56,57,58,59] as well as the cleavage of Angiotensin I and Angiotensin II [60,61], these approaches could lead to major issues in the clinical scenario [57,62,63]. Therefore, TMPRSS2 could represent a valid alternative target in COVID-19 [64,65,66].

Matsuyama and collaborators have shown that TMPRSS2 is expressed in lung tissues and is a fundamental determinant of viral tropism and pathogenicity at the initial site of SARS-CoV infection [67]. Additionally, TMPRSS2 might promote viral spread through a reduced viral recognition by neutralizing antibodies [68,69].

The lung airway expression of both TMPRSS2 and ACE2 was found to be significantly upregulated in smokers compared with non-smokers, and in patients with chronic obstructive pulmonary disease compared with healthy subjects [70]; instead, children were found to have significantly lower expression of COVID-19 receptors in the upper and lower airways (nasal and bronchial) [70]. A recent report analyzing via single-cell sequencing samples obtained from 16 human donors has demonstrated that TMPRSS2 is expressed both in lung tissue and in cells derived from subsegmental bronchial branches, whereas ACE2 is predominantly expressed in a transient secretory cell type [71].

The key role of TMPRSS2 in COVID-19 is corroborated by the observation of an increased TMPRSS2 expression in the bronchial epithelial cells of male patients compared with female patients [72], which could provide an underlying explanation for the previously reported finding of an independent association of male sex with severe COVID-19 [73,74,75].

Of note, TMPRSS2 is a host factor that is also essential for pneumotropism and pathogenicity of the influenza virus [76], thereby suggesting major implications for its direct molecular targeting.

TMPRSS2 is a highly polymorphic gene and some genetic variants of TMPRSS2 have been identified (e.g., rs12329760 and rs75603675), showing that their frequencies vary by geography and ancestry [72,77,78,79,80]. Further studies are needed to determine the effects of miR-98-5p on these variants.

Our study does have some limitations. First, we only conducted in vitro experiments testing the link between miR-98-5p and TMPRSS2 mRNA, without verifying the effects of miR-98-5p on SARS-CoV-2 infection. Second, we focused on TMPRSS2, and we did not investigate the potential contribution of other co-factors needed for the entry of SARS-CoV-2 in the host cells. Since most of the results shown are with exogenously expressed targets or miRNAs, further studies are necessary to appraise the actual translational potential of our findings. The study also has some strengths, including the fact that the 3′-UTR of TMPRSS2 targeted by miR-98-5p is highly conserved among species, from humans to bats (Figure 1). Moreover, we mechanistically validated the effects of miR-98-5p in two different types of human endothelial cells, namely HMVEC-L and HUVEC, obtaining similar results.

## Figures and Tables

**Figure 1 biomedicines-08-00462-f001:**
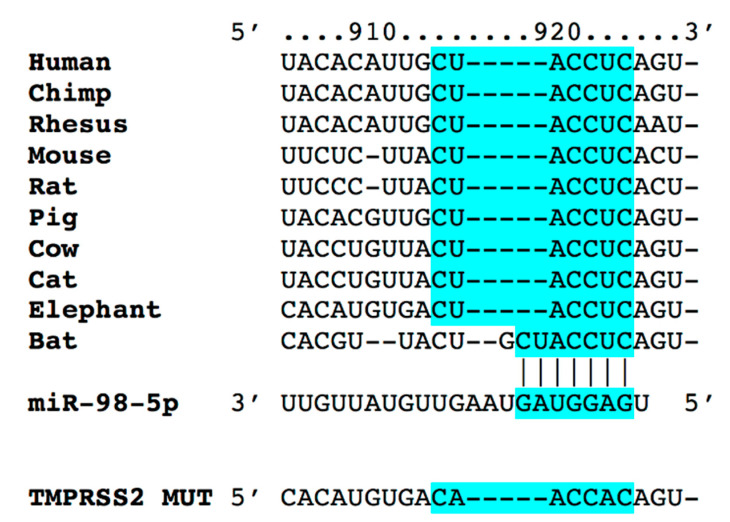
Identification of miR-98-5p as modulator of transmembrane protease serine 2 (TMPRSS2). Complementary nucleotides between the target region of TMPRSS2 3′-UTR (in light blue) and hsa-miR-98-5p are highly conserved across different species.

**Figure 2 biomedicines-08-00462-f002:**
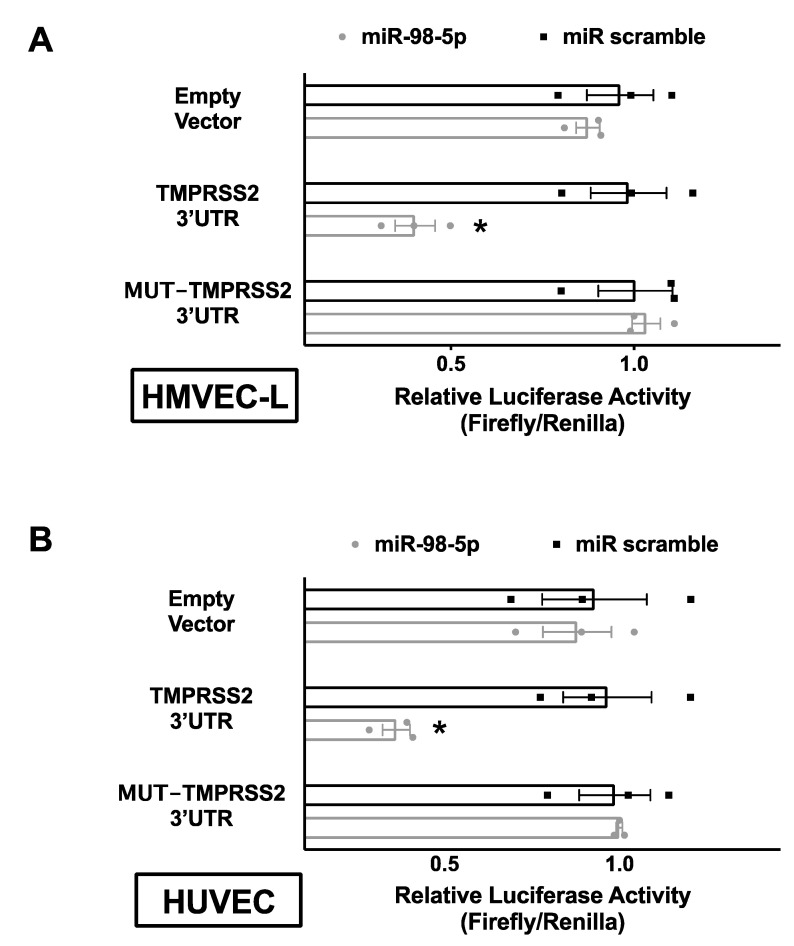
Validation of TMPRSS2 targeting by miR-98-5p. Luciferase activity was measured in adult human lung microvascular endothelial cells (HMVEC-L) (**A**) and human umbilical vein endothelial cells (HUVEC) (**B**) 48 h after transfection, using the vector without TMPRSS2 3′-UTR (empty vector), the vector containing the wild-type TMPRSS2 3′-UTR, and the vector containing a mutated TMPRSS2 3′-UTR (TMPRSS2 MUT); a non-targeting miRNA (miR scramble) has been employed as further control. Means ± S.E.M. are shown alongside actual values; * *p* < 0.05.

**Figure 3 biomedicines-08-00462-f003:**
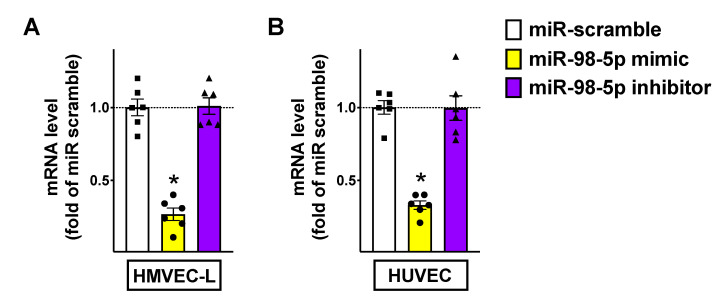
TMPRSS2 expression in human endothelial cells is reduced by miR-98-5p. TMPRSS2 mRNA levels were measured in HMVEC-L (**A**) as well as in HUVEC (**B**) transfected with miR-98-5p mimic, inhibitor, or scramble (negative control) for 48 h. Means ± S.E.M. are shown alongside actual values; * *p* < 0.05 vs. miR scramble.

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
