# Peer review of "miR-98 Regulates TMPRSS2 Expression in Human Endothelial Cells: Key Implications for COVID-19"

_biomedicines, 2020, doi:10.3390/biomedicines8110462_

Round 1

Reviewer 1 Report

This study investigates the microRNAs that specifically target the transmembrane protease serine 2 (TMPRSS2) TMPRSS2, a surface protein on endothelial cells which could bind SARS-CoV-2.

The authors found miR-98-5p 27 as a regulator of TMPRSS2 transcription in two different human endothelial cell types, derived from the lung and umbilical vein.  They concluded that TMPRSS2 represents a valid target in COVID-19 treatment, which may be achieved by specific non-coding RNA approaches.

The present topic of this manuscript appears timely. The main idea of this manuscript and obtained results are interesting and of great interest for the current situation.

Although descriptive, the manuscript is general well assembled, organized and wrote. The data are well and clear presented.

In this context, I recommend that the article be published.

Author Response

Thanks!

Reviewer 2 Report

The paper describes the identification of miR-98-5p as an miRNA that targets the 3´ untranslated region of TMPRSS2. Using a mimic and an inhibitor the authors show that this miRNA regulated the expression levels of TMPRSS2 in two different human endothelial cell lines. As TMPRSS2 is used by COVID-19 to enter bronchial epithelial cells, this finding may have implications for novel approaches to treat the disease.

The findings are new and interesting, and the experimental part convincing. However, only in vitro data are shown, which reduces the impact of the data. Also, whether interfering with miR-98-5p will have an impact on COVID-19 infection is not directly addressed.

Other point: it would be good if the authors discussed the physiological roles of TMPRRS2 and miR-98-5p in lung in more detail.

Author Response

-We thank this Reviewer for the words of appreciation toward our work and for the fast review. As requested, we are now discussing how interfering with miR-98-5p will have an impact on COVID-19. We are also including among the limitations of our study that we are only presenting in vitro data.

-As requested, we are now discussing the physiological roles of TMPRRS2 and miR-98-5p in lung and in endothelial (dys)function.

Reviewer 3 Report

In my opinion the data presented is a bit short. The authors discuss what the drawbacks of the study are, but they don't describe why they haven't done it. 

They should first of all check whether TMPRSS2 is highly expressed in the cell lines they use. As they already suggest, protein levels should be measured in order to check for the biological relevance of the reduction in mRNA levels.

Furthermore it would be interesting to know what is upstream of miR-98-5p. Is the miRNA generally abundant or only abundant under certain conditions. 

It would be also interesting to show at least the top-ranked miRNAs of the bioinformatic screen and an intense justification for choosing miR-98-5p.

Author Response

-We thank this Reviewer for the fast review and for the pertinent comments. As requested, we are now presenting new data in terms of protein levels, as requested. We have also extended the discussion of the drawbacks of the study.

-We now clarify that several reports have previously demonstrated that human endothelial cells express miR-98-5p in basal conditions, and such expression has been shown to be modulated by different stimuli, including hypoxia (Li, H. W.; Meng, Y.; Xie, Q.; Yi, W. J.; Lai, X. L.; Bian, Q.; Wang, J.; Wang, J. F.; Yu, G., miR-98 protects endothelial cells against hypoxia-reoxygenation induced-apoptosis by targeting caspase-3. Biochem Biophys Res Commun 2015;467:595-601) and oxidized low-density lipoproteins (Chen, Z.; Wang, M.; He, Q.; Li, Z.; Zhao, Y.; Wang, W.; Ma, J.; Li, Y.; Chang, G., MicroRNA-98 rescues proliferation and alleviates ox-LDL-induced apoptosis in HUVECs by targeting LOX-1. Exp Ther Med 2017;13:1702-1710). As requested, the 3 top-ranked miRNAs of the bioinformatic screen are now reported in the revised version of the paper, alongside a justification for choosing miR-98-5p.

Round 2

Reviewer 3 Report

I have no further comments.